# Extra-Intestinal Manifestations of Celiac Disease: What Should We Know in 2022?

**DOI:** 10.3390/jcm11010258

**Published:** 2022-01-04

**Authors:** Marilena Durazzo, Arianna Ferro, Isabella Brascugli, Simone Mattivi, Sharmila Fagoonee, Rinaldo Pellicano

**Affiliations:** 1Department of Medical Sciences, University of Turin, C.so A.M. Dogliotti 14, 10126 Turin, Italy; arianna.ferro@unito.it (A.F.); isabella.brascugli@unito.it (I.B.); simone.mattivi@unito.it (S.M.); 2Institute for Biostructure and Bioimaging, National Research Council, Molecular Biotechnology Centre, 10126 Turin, Italy; sharmila.fagoonee@unito.it; 3Unit of Gastroenterology, Città della Salute e della Scienza Hospital, C.so Bramante 88, 10126 Turin, Italy; rinaldo_pellican@hotmail.com

**Keywords:** celiac disease, extraintestinal, diagnosis, clinical presentation, gluten-free diet

## Abstract

Celiac disease (CD) is a chronic, small-intestinal, immune-mediated enteropathy due to gluten exposition in genetically predisposed individuals. It occurs in about 1% of the population and often remains an underdiagnosed condition. This could be due to the fact that the adult population often lacks the classical signs and symptoms of CD, manifesting only atypical symptoms. In this review we analyzed the main extra-intestinal manifestations of CD which include cutaneous and endocrinological disorders, abnormal liver function tests, and neuropsychiatric features. When CD is not diagnosed and therefore is not treated with a gluten-free diet (GFD), it can predispose to severe complications, not only gastrointestinal. Thus, it is important for clinicians to quickly recognize the atypical manifestations of CD, considering that an early diagnosis can significantly impact on a patient’s prognosis.

## 1. Introduction

Celiac disease (CD) is defined as a chronic, small intestinal, immune-mediated enteropathy precipitated by exposure to dietary gluten in genetically predisposed subjects [1,2].

CD may affect individuals of either sex and at any age, albeit the mean age at the diagnosis is 45 years and up to 20% of all patients are diagnosed over 60 years. It is often an underdiagnosed condition especially in adulthood, whereby two-thirds of affected individuals remain long unrecognized [3].

The reason of the delayed diagnosis can be found in the heterogeneous and sometimes vague clinical presentations. In fact, whereas most children manifest the “classical symptomatology”, composed of diarrhea, steatorrhea, or other malabsorption signs, the adult population often does not and the clinical suspicion arises from extra-intestinal manifestations. Among these, some are quite common and well-known as they are directly correlated to absorption alterations and consequent deficiency status (e.g., osteoporosis due to calcium/vitamin D deficit or iron-deficiency anemia). Nevertheless, other possible extra-intestinal manifestations are less frequent and they really constitute the “atypical or non-classic” form of CD [1,4,5]. These include cutaneous disorders, neuropsychiatric features, abnormal liver functions, reproductive abnormalities, and endocrinological manifestations (Figure 1).

Lastly, several individuals may have no symptoms at all, and CD could be diagnosed accidentally [6].

The pathogenesis of CD recognizes the role of genetic factors—almost all patients possess class II human leukocyte antigen (HLA) DQ2 and DQ8, or their variants, even though they are carried by up to 40% of people with European and Asian origins. This finding suggests that these molecules’ expression is necessary but not sufficient for the development of the disease [7].

In the atypical disease forms, diagnosis firstly passes through the ability of physicians of various specialties, including gastroenterologists, internists, pediatricians, neurologists, dermatologists, gynecologists, and particularly general practitioners, to suspect CD from an extra-intestinal manifestation [3]. Secondly, it is necessary to achieve serology, posed by IgA anti-tissue transglutaminases type 2 (anti-tTG2) and total IgA count to exclude an innate IgA deficiency. Several studies demonstrated that in patients with innate IgA deficiency, the first-line strategy is represented by detection of deamidated gliadin peptide IgG antibodies, which have the same performance in the diagnosis of CD as IgA anti-tTG2. [8] Lastly, the suspicion must be confirmed by histopathologic examination of duodenal biopsies, in which the typical inflammatory finding range from lymphocytic enteritis to various degrees of villous atrophy [6].

The aim of this review is to provide a clinical overview of the spectrum of atypical CD signs and symptoms not due to malabsorption or deficiencies. Knowledge of these will help clinicians to carry out an appropriate diagnostic workup and to promptly start the gluten-free diet (GFD). Actually, an early GFD is the only approved life-long treatment to achieve clinical and histological recovery in most patients and to prevent the onset of long-term complications [4].

## 2. Cutaneous and Mucosal Manifestations

There is growing evidence about the predisposition of CD patients to develop several skin disorders, including psoriasis, atopic dermatitis (AD), urticaria, alopecia areata (AA), chronic ulcerative stomatitis, and dermatitis herpetiformis (DH), the only one with a demonstrated gluten-related immune mechanism [1,9].

In patients suffering from CD, the direct gliadin toxicity within the intestinal surface leads to an impaired permeability of the hemato-intestinal barrier, which results in the passage of gluten peptides into the bloodstream. This mechanism is at the root of the aberrant inflammatory and autoimmune activation which features in celiac patients, and which may involve organs and tissues, even extra-intestinal. The inflammatory cascade is activated by the action of tTG2 which catalyzes the structural subversion of gluten, creating new epitopes. These have an increased affinity for HLA-DQ2 e DQ8, which accounts for the susceptibility of these individuals to CD development. The activation of the Th2 response leads to the release of pro-inflammatory cytokines (including tumor necrosis factor [TNF]-α and interferon-γ) involved in the immunopathogenesis of several extra-intestinal manifestations, such as psoriasis. Secondly, the activation of Th1 response can in turn stimulate the proliferation of B lymphocytes which release autoantibodies including those against gluten and tTG2 [4,10].

### 2.1. Dermatitis Herpetiformis

Dermatitis herpetiformis, also called Duhring’s disease, is the most common extra-intestinal manifestation of CD [4,7]. It is characterized by small itchy vesicles and papules symmetrically distributed on extensor surfaces of the limbs, such as elbows, knees, and buttocks, which are often covered by excoriations [9,11].

In Finland and United Kingdom (UK), the prevalence ratio between DH e CD is 1:8 [4,11]. It affects mainly adult males and, in contrast with CD, it seems to be less common in childhood. DH is the typical extra-intestinal manifestation of undiagnosed CD, especially in carriers of HLA-DQ2 or HLA-DQ8 alleles [7,11].

The adjective “herpetiformis” is due to the appearance of the lesions, which are blisters and vesicles gathered into clusters resembling those caused by the herpes simplex virus. Moreover, they are often eroded and crusted owing to the intense itch. The presentation and the trend of the rash is variable and typically affects, as reported above, the extensor surface of the limbs, abdomen, upper back, scalp, and face, while the involvement of feet and hands is rare [12].

Biopsy of unaffected skin, in the proximity of an active lesion, is necessary for diagnosis. The typical histopathological finding consists of subepidermal vesicles with neutrophilic micro-abscesses in the papillary tips. Immunofluorescence instead reveals granular or fibrillar IgA (and C3) deposits at the dermo–epidermal junction [9,11]. Nowadays, the most accepted hypothesis on the immune pathogenesis of DH is that this disease begins in the gut of patients with hidden CD, which starts IgA anti-tTG production (especially type 2). Subsequently, anti-tTG3 antibodies are produced, in which type 3 transglutaminase is the epidermal antigen, developing an immune-complex deposition disease in the papillary dermis. Accordingly, it has been observed that tTG3 and tTG2 antibodies disappear from the blood as well as the rash, after a GFD. The IgA–tTG3 aggregates in the skin, instead, disappear slower [11].

Several patients with DH have undiagnosed enteropathy, while only a few of them report gastrointestinal symptoms [1,9]. In fact, when duodenal biopsies are performed in people with DH, a variable degree of villous atrophy is observed in almost 75% of patients, predominantly of a mild to moderate degree; in the remaining 25% only inflammatory signs as in lymphocytic enteritis are visible, without evidence of villous atrophy [4].

DH has an excellent long-term response to a GFD, which is considered its life-long treatment of choice, regardless of the presence of duodenal findings. Sometimes, the response to diet may be slow: that is why most patients initially take dapsone (4,4-diaminodiphenylsulfone), which can be suspended after a mean of two years of a rigorous GFD [11]. If DH persists, despite a GFD, an ongoing gluten exposure must be suspected [9].

### 2.2. Urticaria

Urticaria is a condition that occurs with wheals, angioedema, or both. It is defined “chronic” (CU) when the duration of the disease is longer than 6 weeks and it involves about 0.5–1% of the general population [4]. CU is associated with various autoimmune diseases, including pernicious anemia, thyroid disorders such as Hashimoto’s thyroiditis or Grave’s disease, vitiligo, type 1 diabetes mellitus, rheumatoid arthritis, and last, but not least, CD. It has been reported the genetic association of CU with the HLA-DQ8 alleles [7]. As in other extraintestinal manifestations of CD, in several cases the adoption of a GFD is able to control skin flares, showing that CU is a cutaneous expression of CD and not only an occasional association [9].

### 2.3. Atopic Dermatitis

Atopic dermatitis is a chronic inflammatory skin disease which occurs four times more frequently in CD patients than in general population [1]. AD is characterized by itch and pain, and the skin lesions include erythema, lichenification, and scaling. This disease is more common among children under five years of age, and its prevalence diminishes with advancing age. However, available data concerning the efficacy of a GFD in atopic patients with CD are still insufficient [4].

### 2.4. Psoriasis

Psoriasis is a chronic inflammatory skin disease which causes a severe reduction in patients’ quality of life. It is characterized by red, infiltrated, and well-demarked plaques, covered with a coarse and silvery scales usually located in the elbows, knees, scalp, and periumbilical and lumbar regions, albeit every anatomical site might be involved. The disease presents a fluctuating trend, with exacerbating and remitting periods [7]. The association between psoriasis and CD has been reported since 1997, but was recently re-investigated by Bhatia et al. [13]. These authors performed a meta-analysis of nine studies analyzing the frequency of IgA anti-gliadin antibody (AGA) positivity in patients with psoriasis and controls. They found a significantly higher relative risk of positive IgA AGA in psoriatic patients than in controls (odds ratio (OR) = 2.36, 95% confidence interval (CI) 1.15–4.83) [13]. Moreover, a correlation between CD antibody titer and the severity of psoriatic lesions was demonstrated [7].

Several hypotheses have been formulated to explain the association of these two diseases—firstly, the shared genes such as HLA-haplotypes might play a role similar to that in other diseases with an autoimmune background, including type I diabetes mellitus and autoimmune thyroid disease. Secondly, the keratinocyte proliferation that characterizes psoriasis could produce an excess of interleukin (IL)-1 and IL-18, which, in turn, induces activation of the Th1 response, considered one of the agents responsible for the mucosal inflammation in CD. Another possible explanation includes the impaired gastrointestinal barrier of the undiagnosed or untreated CD, which may facilitate the passage of antigens causing an immune trigger and consequently an autoimmune activation. Finally, vitamin D deficiency status owed to CD-related malabsorption could predispose individuals to psoriasis, given also that sunlight exposure or the topical application of vitamin D analogues can induce the remission of psoriatic lesions [4].

Most recent data suggest that in patients with psoriasis and CD, a GFD may improve both the diseases, especially in those with a high titer of IgA AGA. For this reason, it is useful to test for CD in psoriatic patients, to identify the ones who are likely to benefit from a GFD [7,9].

### 2.5. Rosacea

Another cutaneous manifestation which shows an increased risk in CD patients, especially women, is rosacea. This is an inflammatory disease characterized by persistent erythema in the region of cheeks and nose which can seriously worsen patients’ quality of life. Although its pathophysiology is not completely known, the predominant mechanism is the immune dysregulation, which explains the increased prevalence reported by some studies in patients with inflammatory bowel disease (IBD), irritable bowel disease (IBS), small intestinal bacterial overgrowth, or *H. pylori* infection [4].

### 2.6. Oral Manifestations

The oral cavity is one of the most-affected areas in CD patients, appearing with enamel defects (ED), delayed dental eruption, and recurrent aphthous stomatitis (RAS).

Bramanti et al. in 2014, reported that ED was more frequently observed in children with typical gastrointestinal manifestations, while aphthae were more prevalent in silent and undiagnosed CD [14]. Dental ED associated with CD are specific—they involve all the four quadrants, with color and enamel surface alteration following the sequence of dental mineralization [5,14,15]. Other ED, such as discolorations or opacities, especially if not symmetrical nor chronological, are considered unspecific [16].

Dental ED in CD patients can be classified according to Aine’s classification in four degrees—from enamel color imperfections (grade I) to severe structural defects (grade IV) (Table 1) [16].

As mentioned before, CD patients often show RAS. This is a condition characterized by several oral aphthae, burning sores which may appear in non-keratinized oral mucosa and typically concern childhood or adolescence. They are usually round or ovoid in shape, surrounded by an erythematous edge with a yellow or gray bottom [4,16]. These lesions affect feeding, speech, or swallowing and cause considerable pain [16]. Nieri et al. performed a quantitative meta-analysis to compare dental ED and aphthae in healthy people versus CD patients; they demonstrated that the latter may be affected by aphthous stomatitis three times more than healthy controls, with a prevalence approximately of 50% versus 10–20% of the general population [6,14,16].

There are multiple mechanisms responsible for oral manifestations in CD, which involve the direct effect of gluten sensitivity disorder or the consequences of malabsorption, with iron, folic acid, and vitamin (A-D-B12) deficiency [4,15,17]. Moreover, the celiac enteropathy induces an alteration of phospho–calcium metabolism, causing hypocalcemia and consequent dental demineralization [15].

Concerning dental ED, the presence of similar sequences between gliadin and dental enamel proteins recognized by CD antibodies has also been reported [14]. Moreover, as in other autoimmune diseases, the HLA alleles seem to play a role in protection/promotion of oral symptoms’ development. For example, the HLA DQB1 has a protective role for ED and RAS in CD, while the HLA DR52-53 alleles are more commonly observed in children with alteration in dental enamel [14]. On the other hand, RAS can rely on many other factors such as family history, local trauma, stress, hormonal imbalance or immune changes, IBD, food hypersensitivity, and nutritional deficiencies [15,16]. Nevertheless, people manifesting RAS should be considered at-risk subjects for CD development despite the absence of any gastrointestinal symptoms and they should undergo the diagnostic process for CD exclusion [4,6]. In fact, RAS caused by CD typically regresses with a strict GFD [14].

Other oral manifestations that CD children can manifest as a result of a prolonged malabsorption or malnutrition are delayed tooth eruption, geographic tongue, atrophic glossitis, and angular cheilitis [16]; xerostomia is also frequent. One of the known causes of insufficient saliva production in CD patients is Sjögren’s syndrome and in these cases a GFD has no beneficial influence on symptomatology [17].

### 2.7. Alopecia Areata

Alopecia areata is a form of autoimmune non-scarring alopecia in which clinical manifestations may vary from small and well-delimited regions of hair loss to the complete absence of scalp hair. It is quite a common disease in general population (prevalence is about 2%), affecting all sexes and ages. The diagnosis makes use of trichoscopy and is indicated from signs such as exclamation point hairs, dystrophic hairs, and yellow dots [4,9].

AA is considered a disimmune disorder, characterized by the loss of immunotolerance for hair follicles, resulting in follicle destruction [4]. The association between CD and AA has been demonstrated by several studies. These showed an increased prevalence of AGA in patients with AA, especially in the severe forms [18,19]. Moreover, the introduction of a GFD may induce a significant reduction in alopecia severity and recurrence [18].

### 2.8. Cutaneous Vasculitis

Several studies analyzed the association between CD and cutaneous vasculitis, especially the leucocytoclastic one. The latter, also named hypersensitivity vasculitis, is a small vessel vasculitis caused by the circulating immunocomplex deposition into the vessel walls, with the activation of a complement pathway. It usually occurs with palpable purpura, hemorrhagic bullae, papules, nodules, or ulcers, and is accompanied by elevated gluten-fraction antibody titer [7].

Few papers, mainly case-reports, about cutaneous vasculitis and CD association have been published in the literature, including the most recent one about a 38-year-old woman with untreated CD and leucocytoclastic vasculitis who had complete remission of the skin lesions after the adoption of a strict GFD [7,9].

### 2.9. Other Cutaneous Manifestations

Other cutaneous manifestations described in case-series on CD patients are pemphigus, hereditary angioneurotic edema, erythema nodosum, erythema elevatum diutinum, necrolytic migratory erythema, vitiligo disease, oral lichen planus, dermatomyositis, porphyria, acquired hypertrichosis lanuginose, pyoderma gangrenosum, ichthyosiform dermatoses, pellagra, generalized acquired cutis laxa, and skin malignancies. Since the causal association between these lesions and CD has not yet been established, these case-series are not detailed in this review [7].

## 3. Neurological Manifestations

The relationship between CD and neurologic disorders was first described in 1966, by Cooke and Smith [20], who observed evidence of cerebellar ataxia and peripheral neuropathies in some CD patients. Nowadays, although gluten neuropathy and ataxia remain the most common [21], many other neurologic disorders have been correlated to CD, such as headache, epilepsy, and cognitive impairment [22].

Overall, it has been estimated that one-fifth of CD patients suffer from neurological manifestations [3]. A recent study by Hadjivassiliou et al. [23] observed that at the time of diagnosis, 67% of patients already have signs of neurologic dysfunction.

However, the precise pathophysiological process of neurological involvement in CD remains partially unclear. Certainly, the nervous system undergoes many gluten-mediated mechanisms, including cross-reacting antibodies, immune-complex deposition, direct toxicity and gut–microbiota–brain axis alteration [24]. In recent years, some authors have also hypothesized that neurologic disorders could be a consequence of an abnormal brain perfusion [22,25]. Using single-photon emission computed tomography, they noted the presence of cerebral hypoperfusion in CD patients, which might be related to intestinal hyperemia and/or perivascular inflammation [22].

### 3.1. Ataxia

Cerebellar ataxia (CA), also called gluten ataxia, is one of the first neurologic symptoms and one of the most frequent in patients with CD [20]. A recent systematic review estimated the prevalence of gluten ataxia to be 0–6% among CD patients and observed that ataxia accounts for a higher percentage of neurological manifestations in CD adults compared to children [22].

CA could be the only and first clinical manifestation of CD. It usually appears in late mean age (55 years old) and without any other associated gastrointestinal symptoms [26].

In most cases (69%), symptoms are mild and include difficulty with arm and leg control, altered movements of eyes and poor coordination. However, in one-third of patients CA leads to a progressive impairment of stability with moderate/severe gait manifestations which require walking support or even wheelchair use [27].

From a histological point of view, patients with gluten ataxia show a particular type of neurologic deficit consisting of the loss of Purkinje cells. This damage was firstly attributed to vitamins deficiencies, such as B1, B3, B6, and B12, which were well-known causes of neurological disorders [28]. However, later findings demonstrated an immune-mediated pathogenesis [28]. In fact, AGA and anti-tTG (particularly tTG6) have shown reactivity with the deep cerebellar nuclei brainstem and cortical neurons, which bring to a cross-reaction with Purkinje cells and their consequent damage [21,23,28]. So, it is not surprising that the prevalence of circulating AGAs and/or tTG6 antibodies was found to be higher in CD patients with CA rather than in those without neurological problems (73% vs. 40%) [23].

Diet seems to be effective on gluten ataxia—the higher the adherence to a strict GFD, the better are the improvements in cerebellar function and clinical manifestations [28]. These effects are probably due to the positive impact of gluten exclusion on autoimmunity mechanisms. However, ataxic symptom relief with a GFD has been proven only in adult CD patients. Hence, its usefulness is unclear among children [21].

### 3.2. Peripheral Neuropathy

Peripheral neuropathy is the second-most-common neurological manifestation reported in CD patients [20,21]. Its prevalence ranges from 2% to 23%, with higher rates among females and older patients [21,23,29].

The presence of peripheral neuropathy is often discovered from mild sub-clinical manifestations such as lower sensibility to pain threshold, warm or skin contact, feeling of numbness or tingling, and writing difficulty [20,30]. It is usually described as a symmetric neuropathy, so it involves both the arms and legs, and it can progressively lead to gait instability.

As a characteristic symptom does not exist, a skin or nerve biopsy is needed to confirm the diagnosis. Unlike gluten ataxia, in the case of neuropathy histological examination highlights the loss of myelinated fibers [30].

The effect of a GFD on peripheral neuropathy is still unclear. Some studies suggest that a GFD could improve nerve function and symptoms. However, diet adherence seems not to prevent neuropathy development and the severity of the disease is not associated with GFD duration [31].

### 3.3. Headache, Epilepsy, and Cognitive Impairment

Other neurological manifestations correlated to CD are headache, epilepsy, and cognitive impairment [22].

In the early 2000s, Cicarelli et al. observed a higher prevalence of headache among CD patients and used, for the first time, the term “gluten encephalopathy” [32]. This condition is characterized by episodic headache attacks, which are similar to migraine and affect about 26% of celiac population [33].

From a histological point of view, CD patients with headache show some white manner alterations at magnetic resonance neurography (MRN). These lesions seem to be correlated not so much to demyelination as to a vascular inflammation process [30].

Adherence to a GFD is effective for symptom improvement and leads to total resolution of headaches in up to 75% of patients. The precise mechanism of this effect remains unclear, but it is probably due to inflammation reduction. Nevertheless, a GFD is not able to resolve white matter abnormalities [33,34].

Instead, Pfaender et al. collected a series of cases in which children with biopsy-confirmed CD had epilepsy and bilateral occipital calcifications [35]. This association was also described by other authors [36,37], who suggested that HLA genotypes predisposing to CD could be the same as those predisposing to bilateral occipital calcifications and epilepsy [35].

The risk of epilepsy in CD population is increased of 1.4 times, with a prevalence of 1–6%. However, neither the etiopathogenesis nor the correlation between CD and epileptic crisis are completely understood [38]. Some researchers have associated the presence of cerebral calcifications to a gluten-toxicity process [39,40].

Manifestations of CD crisis are very similar to those of epileptic syndrome, including blurred vision and colored dots’ view, and they are usually well controlled by antiepileptic medications [30].

The effect of a GFD is evident in most patients, in whom it determines better seizure control, decreased drug use, and even complete resolution. Moreover, a GFD seems to arrest cerebral calcification development [36,39].

CD can also cause cognitive function damage. The presence of cognitive impairment in celiac patients was first reported by Kinney et al. [41]. Since then, many studies have confirmed the correlation between CD and cognitive impairment, but its real prevalence remains difficult to define. In fact, symptoms are non-specific and variable, in both severity and duration. Usually, they are mild and transient, and include difficulties in concentration, inattention, struggling to find words, and episodic memory deficits. The condition of slight cognitive alteration, that is known as “brain fog”, is the most common symptom among CD patients [42]. However, in some cases, cognitive impairment gets worse and may lead to confusion, disorientation, and even dementia [42].

Several mechanisms have been proposed to explain the gluten-induced deleterious process. Nowadays, two hypotheses are the most accredited. The first attributes cognitive damage to systemic inflammation caused by CD, which also favors cerebral inflammation and consequent reduction in neuronal transmission speed. The second blames gluten for indirectly reducing brain serotonin levels [42,43].

From a histological point of view, patients with cognitive impairment do not show particular radiological lesions; at most, diffuse cerebral atrophy can be observed. For this reason, diagnosis can be performed only through neuropsychological tests [42].

A GFD is effective for brain fog and other mild alterations, which often completely disappear. However, its effect on more severe forms remains controversial [44].

## 4. Neuropsychiatric Manifestations

The relationship between CD and psychiatric disorders is known, but it is not yet fully recognized and understood. Depression, anxiety, attention deficit/hyperactivity disorder (ADHD), and autism are the most frequent neuropsychiatric manifestations in celiac patients [45].

The precise pathophysiological mechanism of gluten-induced psychiatric involvement is unclear. However, both biological and social factors might be involved. Biological explanations include inflammation processes, autoimmunity activity, microbiota composition, and the gut–brain axis relationship, but evidence in this regard is scarce especially due to small sample sizes of the studies [45]. Instead, social implications are easier to guess. They include all the possible negative consequences of GFD introduction, such as social isolation or avoiding going out because of difficult meal management and contamination risk [45,46].

### 4.1. Depression

Smith et al. observed that depressive disturbance was more common and more severe among adult CD subjects [47]. According to a recent systematic review, the risk of depression development is higher in this cohort, with a mean prevalence of 3.5% [45].

The increased risk has been explained by various biological hypotheses, such as low levels of serotonin due to tryptophan malabsorption, concomitant presence of hypothyroidism, and increased hypothalamic–pituitary–adrenal axis activity [45,46]. On the other hand, adherence to a strict dietary regimen can induce patients to avoid social situations involving food, and to have higher levels of stress because of meal difficulties and worries. All these psycho–social conditions certainly contribute to depression onset and progression [46].

As diet stress is one of the possible contributing causes of depressive manifestations, the role of a GFD is controversial. Some studies observed improvement with GFD adherence, especially after long-term administration [48], while others even reported the worsening of depressive symptoms [45,49].

### 4.2. Anxiety, ADHD, and Autism

Anxiety states are more common among new-diagnosed CD patients. In fact, the introduction of a GFD leads the subject to make many changes in eating habits and lifestyle, which can result in agitation and feelings of stress [46]. However, this condition usually ameliorates after 1 year of GFD [49].

ADHD affects about 1.4% of CD patients, whose risk is higher compared to the general population [46]. A preliminary study by Niederhofer et al. found an “overexpression” of ADHD symptoms in a group of celiac subjects, which improved in the majority of them (74%) after 6 months of a GFD [50].

A significant increase in risk development has also been observed for autistic spectrum disorders, in which an appropriate compliance to a GFD seems to improve behavioral symptoms [46].

Although the correlation of anxiety, ADHD, autism, and CD is often described, precise etiological mechanisms and specific biological explanations of GFD effectiveness remain unclear and need further investigation.

### 4.3. Other Psychiatric Disorders

Evidence concerning correlation between CD and other psychiatric disorders is very limited and mainly published as case reports.

Some researchers recently hypothesized that CD patients are at higher risk of having eating disorders. However, only a few studies have investigated the impact of a GFD on them [45].

Many surveys have evidenced symptom improvement in schizophrenic subjects after the introduction of a GFD. This has stimulated scientists to investigate the possible association between CD and schizophrenia, also through research on genetic similarities [51,52]. However, recent studies challenged this theory and showed no significant correlations [45].

Likewise, no significant differences in prevalence and risk have been observed for bipolar disorder [45].

## 5. Liver Manifestations

It is not rare that CD is included in the history of illness signs of hepatic injury.

Liver damage can be due to a dysmetabolic condition, such as non-alcoholic fatty liver disease [53], or to a concomitant autoimmune disease, such as autoimmune hepatitis [54]. However, recent studies showed elevated aminotransferase levels in celiac patients without known causes of liver disease [55,56]. This condition has been called “gluten-induced hepatitis”, with a prevalence of 4–9% reported by recent studies [57].

Usually, hepatic injury is mild and easily reversible, leading to liver failure on rare occasions. Behind the physio-pathological mechanism of the hepatotoxicity, there is the alteration of gut permeability, which leads to an increased exposure to hepatotoxins in the portal circulation [55]. Moreover, a few studies have reported histological evidence of tTG2 autoantibody deposits in liver biopsies from CD patients, suggesting their direct involvement in hepatocellular damage [3].

So, the search for CD should be performed in subjects who present no symptoms of chronic liver disease nor other reasons for high transaminase levels [55]. The severity of the hypertransaminasemia has been associated with the presence of malabsorption, high titer of celiac autoantibodies and severe duodenal lesions [12]. When a liver biopsy is performed, generally in patients who show persistently high liver enzyme levels, a minimal grade steatosis or a reactive nonspecific hepatitis can be observed [55].

In celiac patients, liver condition is generally benign and reversible—GFD usually leads to injury resolution and transaminases normalization within 6–12 months [57].

Since it may often occur as an isolated hypertransaminasemia in the absence of other gastrointestinal symptoms, it must be included in hepatitis differential diagnosis [56]. Only where CD is unrecognized or untreated and the liver involvement is subclinical, is progression of the hepatic involvement to be feared. Surprisingly, as a few studies have demonstrated, GFD allows a clinical improvement even in these more severe conditions [56,58].

Therefore, CD has to be carefully investigated in all patients with hepatitis of unknown etiology, since early detection and treatment can prevent the progression to severe liver disease [55].

## 6. Reproductive Manifestations

It is known that CD can cause inflammation and malabsorption. Both these conditions might be responsible, or at least partially contribute, to infertility [59]. However, some researchers have also proposed the possible role of autoimmunity, suggesting that anti-tTG antibodies may be responsible for the inhibition of endometrial angiogenesis [60]. Whatever the cause, GFD seems to be effective. In fact, celiac women undergoing a GFD have a fertility which is comparable to healthy ones [61,62], whereas women who do not follow a GFD can manifest several conditions such as late menarche, secondary amenorrhea, early menopause, miscarriages, preterm pregnancy, and low birth weight of the newborn [62,63,64,65]. Specifically, the risk of recurrent miscarriages is eight to nine times higher in celiac women who do not follow a GFD than in the general population [66]. As observed by Fortunato et al. [67], prevalence of reproductive disorders is higher in patients with CD compared to the general population, suggesting how it may be useful to test for CD in women presenting changes during pregnancy or infertility [67].

Thus, CD can be considered as a risk factor for infertility. The favorable response to a GFD can justify the serological screening for CD among women with infertility of unknown origin [65,68]

Nevertheless, not all the studies agree with this theory. In a recent meta-analysis, the authors reported that using a more stringent definition of CD (confirmed by biopsy requiring a Marsh type III villous atrophy), the prevalence of this condition in women with infertility should be similar to that observed in the general population [68]. This stands in contrast to the meta-analyses by Castano et al. and Singh et al. who instead found a higher prevalence of infertility in patients with CD [69,70]. One of the reasons why this difference can be observed may be found in the definition of CD—some studies have considered Marsh type I as sufficient for the diagnosis of disease while others have not even used Marsh’s criteria for diagnosis. In the meta-analysis by Glimberg et al., CD diagnosis had to be verified by a biopsy that confirmed a Marsh type III and excluded couples in which the underlying problem was male infertility. In a second analysis, studies were evaluated in which, for the definition of CD, the presence of IgA tTG was sufficient [71]. However, the authors highlight how the lack of fertile women used as control can be a limitation as well as the low number of participants in the majority of the studies [71]. Further cohort studies are needed to better understand the correlation between CD and infertility.

Focusing on the male population, sperm abnormalities both in terms of morphology and motility can be observed in CD patients [72,73]. From a strictly biochemical point of view, some studies reported a condition of androgen resistance, which means high testosterone levels and high LH values [72,73]. However, androgen resistance seems to respond to a GFD. In fact, a study conducted on 41 CD males with androgen resistance showed that a GFD determined the normalization of biochemical alterations [73].

## 7. Endocrinological-Associated Diseases

While cutaneous, neuropsychiatric, hepatic, and reproductive manifestations mostly regress or even disappear following a strict GFD, endocrinological alterations do not. For this reason, the latter are typically described as “associated diseases” of CD.

These disorders are usually characterized by a mono-glandular involvement (e.g., thyroid), but they can also be polyglandular [74]. Diagnosis of CD may also lead to definition of a single or multiple gland autoimmune syndrome [75]. This highlights the importance of awareness and regular screening for these complications in patients with CD and vice versa [76].

### 7.1. Type 1 Diabetes Mellitus and Autoimmune Thyroiditis

Type 1 diabetes mellitus (T1DM) and autoimmune thyroiditis (AT) are the most common endocrinopathies among celiac patients. The former usually affects children and adolescents [77,78] while the latter is more frequent among adults [79,80].

The common pathophysiological process, based on an autoimmune response, makes the association of T1DM and AT with CD quite easy for clinicians, who often request celiac screening in this subgroup of patients.

### 7.2. Autoimmune Polyendocrine Syndrome

In many CD patients, an autoimmune polyendocrine syndrome (APS) can also be observed [81]. APS type I often develops in adolescence and leads to multiple endocrinology deficits such as mucocutaneous candidiasis, skin dystrophy, and various endocrine disorders such as hypoparathyroidism, Addison’s disease, T1DM, hypogonadism, and thyroiditis. On the other hand, APS type II develops around the third of fourth decade of life, is more common in females, and is characterized by AT, T1DM, and Addison’s disease, with hypoparathyroidism being less frequent and muco-cutaneous candidiasis absent [82].

Although hypoparathyroidism has been rarely recorded with coincident CD [83], a recent report highlighted the beneficial effects of a GFD on calcium regulation [84].

In a Swedish national registry study [85], people with CD, both children and adults, had a positive association with Addison’s disease. It was found that there was a statistically significantly positive association between CD and subsequent AD (HR = 11.4; 95% confidence interval (CI) = 4.4–29.6) [85]. It was recommended that patients with adrenal insufficiency, especially those non-responders to substitute hormonal therapy, should be screened for CD. Also, CD patients should be investigated for adrenal insufficiency specially if associated with recurrent hypoglycemia [82,85,86].

Another form of APS, the type 3, may be seen when there is no adrenal cortical defect. A further type 4 APS may occur if an autoimmune hypophysitis develops [87]. An Italian study, including children and adolescents, showed a high detection rate (42%) of anti-pituitary antibodies in newly diagnosed CD patients [88]. High antibody levels were associated with height impairment, probably due to a reduction in insulin-like growth factor, suggesting that an autoimmune pituitary process could be important in the induction of linear-growth impairment in CD [88]. Another evidence of the association between CD and pituitary gland alteration arises from the prolactin level, which is increased in recently diagnosed CD pediatric patients [89]. In this case, prolactin values decreased over a few months of following a GFD [90].

### 7.3. Short Stature

Short stature is a common extra-intestinal manifestation of CD which, unlike other endocrinological alterations, is sensitive to a GFD [12]. Early diagnosis and treatment among celiac children is associated with early catch-up growth for the initial 2–3 years with the possibility of achieving normal height in adulthood [91]. Malabsorption secondary to the villous atrophy is likely the contributor to this condition, but it is not possible to exclude that the inflammatory process, especially the elevation of pro-inflammatory cytokines (IL-6, TNF-α, and IL-1), can lead to a malfunction in growth hormone secretion [89]. Failure to identify CD has been associated with shorter heights in adulthood as compared with that in normal population [92].

Van Rijn et al. reported that in 2–8% of patients with short stature without prior endocrinological evaluation, CD was the cause of their condition [93].

These data were confirmed by a recent systematic review including 3759 patients, in which prevalence of CD confirmed by a biopsy ranged from 7.4% to 11.6% for all-cause and idiopathic short stature [94]. Therefore, it is important to screen patients with short stature, both children and adults, for CD.

## 8. Conclusions

In conclusion, extra-intestinal manifestations of CD are not so rare. These may include various symptoms and signs, which need to be known and quickly recognized by clinicians, considering that a prompt diagnosis has a significant prognostic impact on patients.

## Figures and Tables

**Figure 1 jcm-11-00258-f001:**
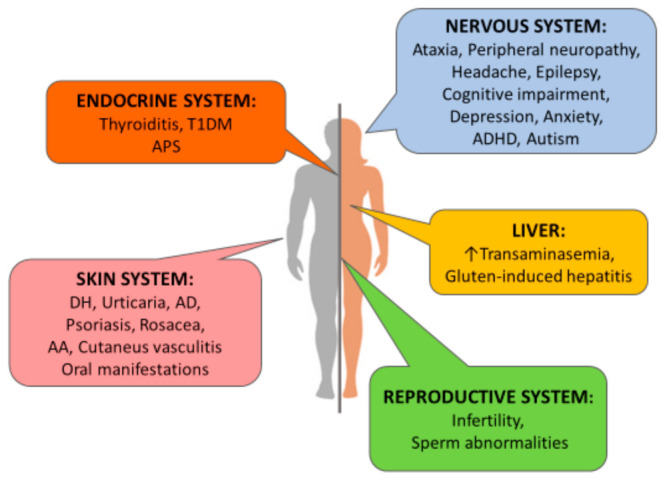
Extra-intestinal manifestations of celiac disease and involved systems. T1DM: Type 1 Diabetes Mellitus; APS: Autoimmune Polyendocrine Syndrome; DH: Dermatitis herpetiformis; AD: Atopic dermatitis; AA: Alopecia areata; ADHD: Attention deficit/hyperactivity disorder.

**Table 1 jcm-11-00258-t001:** Aine’s Classification.

Grade 0	Grade I	Grade II	Grade III	Grade IV
No defects.	Defect in enamel color (yellow or brown marks)	Slight structural enamel defects (rough surface, groves)	Evident structural defects (deep groves, large opacities)	Severe structural defects (lesions)

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
