# Peer review of "Extra-Intestinal Manifestations of Celiac Disease: What Should We Know in 2022?"

_jcm, 2022, doi:10.3390/jcm11010258_

Round 1

Reviewer 1 Report

In the present review, the authors have focussed on certain selected extra-intestinal manifestations of celiac disease. 

Comments: 

1. The authors may like to review update the section on liver manifestations of celiac disease:

Haggård L, G et al. Liver Int . 2021 Nov;41(11):2693–702. 

Sainsbury A, Aliment Pharmacol Ther. 2011; 34: 33-40

2. There are systemic reviews on many endocrinological manifestations such as type I diabetes, thyroid disorders, short stature. Authors may like to add to the text. 

Singh AD, J Gastroenterol Hepatol. 2021 Jan;36(1):44–54.

Elfström P, Aliment Pharmacol Ther. 2014 Nov;40(10):1123–32.

3. Authors may also like to discuss and add fatty liver and metabolic syndrome. 

Author Response

We thank the Reviewer for the comments and suggestions which give us the opportunity to improve our manuscript.

Please find enclosed the revised version of our article, with changes tracked according to requirements of the Journal.

________________________________________________________________________________

In the present review, the authors have focused on certain selected extra-intestinal manifestations of celiac disease. Comments:

  1. The authors may like to review update the section on liver manifestations of celiac disease: Haggård L, G et al. Liver Int. 2021 Nov;41(11):2693–702; Sainsbury A, Aliment Pharmacol Ther. 2011; 34: 33-40

________________________________________________________________________________

As suggested, we updated the section on liver manifestations of celiac disease adding new references (lines 422-453, changes highlighted in light blue, ref. 55,58)

________________________________________________________________________________

  1. There are systemic reviews on many endocrinological manifestations such as type I diabetes, thyroid disorders, short stature. Authors may like to add to the text. Singh AD, J Gastroenterol Hepatol. 2021 Jan;36(1):44–54; Elfström P, Aliment Pharmacol Ther. 2014 Nov;40(10):1123–32.

____________________________________________________________________________

As suggested, in the new version we included paragraphs on Type 1 Diabetes Mellitus, Autoimmune Thyroiditis and Short Stature.  New references have been also added (lines 502-508; 537-552, changes highlighted in light blue, ref. 86,95)

_____________________________________________________________________________

  1. Authors may also like to discuss and add fatty liver and metabolic syndrome.

_____________________________________________________________________________

We included a part focusing on fatty liver disease and other dysmetabolic hepatic conditions (lines 424-426, changes highlighted in light blue, ref. 54). This topic has not been thorough as metabolic syndrome and other correlated conditions usually appear later, after celiac disease diagnosis. “Increases in the frequency of NAFLD, weight gain and alterations of the lipid profile suggest that important changes happen in celiac patients on a GFD…” (Valvano, M. et al. Celiac disease, gluten-free diet, and metabolic and liver disorders. Nutrients 2020).

Reviewer 2 Report

This work is a review of the main extra-intestinal manifestation of CD, it is well written and understandable.

However there are several extra-digestive manifestations that are not mentioned and I consider they should be such as Bone mineral density alterations, Arthritis and hematological diseases like ferropenic anemia or hyposplenism.

Regarding endocrinological manifestations they should be considered as associated diseases rather than extra-digestive CD manifestations

Abbreviation for anti-transglutaminase antibody is written differently in different parts of the tests and has to be checked.

Author Response

We thank the Reviewer for the comments and suggestions which give us the opportunity to improve our manuscript.

Please find enclosed the revised version of our article, with changes tracked according to requirements of the Journal.

________________________________________________________________________________

This work is a review of the main extra-intestinal manifestation of CD, it is well written and understandable.

However, there are several extra-digestive manifestations that are not mentioned and I consider they should be such as bone mineral density alterations, arthritis and hematological diseases like ferropenic anemia or hyposplenism.

________________________________________________________________________________

We agree with the Reviewer comment. In our review we decided to focus on manifestations that are not directly correlated to malabsorption and consequent deficiency status (e.g., osteoporosis due to calcium/vitamin D deficit or iron deficiency anemia). In fact, our aim was to provide a complete clinical overview on the spectrum of atypical celiac disease signs and symptoms, in order to help clinicians in the diagnostic process (lines 33-44; 67-70, changes highlighted in yellow).

___________________________________________________________________________

Regarding endocrinological manifestations they should be considered as associated diseases rather than extra-digestive CD manifestations.

________________________________________________________________________________

We agree with the Reviewer comment and we specified this concept it in the new version of the manuscript (lines 493-496, changes highlighted in yellow).

_______________________________________________________________________________

Abbreviation for anti-transglutaminase antibody is written differently in different parts of the tests and has to be checked.

________________________________________________________________________________

We apologise for the mistake. We standardized the abbreviation throughout text (changes highlighted in green).

Round 2

Reviewer 1 Report

Thank you very much for responding to the comments 

Reviewer 2 Report

I agree with the changes made.